# Effect of Dietary Mineral Content and Phytase Dose on Nutrient Utilization, Performance, Egg Traits and Bone Mineralization in Laying Hens from 22 to 31 Weeks of Age

**DOI:** 10.3390/ani11061495

**Published:** 2021-05-21

**Authors:** Mehran Javadi, Juan José Pascual, María Cambra-López, Judit Macías-Vidal, Andrés Donadeu, Javier Dupuy, Laura Carpintero, Pablo Ferrer, Alba Cerisuelo

**Affiliations:** 1Institute for Animal Science and Technology, Universitat Politècnica de València, Camino de Vera s/n, 46022 Valencia, Spain; m.javadi2012@yahoo.com (M.J.); macamlo@upvnet.upv.es (M.C.-L.); 2Departamento de I+D+i, Global Feed S.L.U., Grupo Tervalis, Av. Francisco Montenegro s/n, 21001 Huelva, Spain; judit.macias@tervalis.com (J.M.-V.); andres.donadeu@tervalis.com (A.D.); javier.dupuy@tervalis.com (J.D.); laura.carpintero@tervalis.com (L.C.); 3Centro de Investigación y Tecnología Animal, Instituto Valenciano de Investigaciones Agrarias, 12400 Segorbe, Castellón, Spain; ferrer_pabrie@gva.es (P.F.); cerisuelo_alb@gva.es (A.C.)

**Keywords:** laying hen, phytase, age, dose, digestibility, egg quality, mineralization

## Abstract

**Simple Summary:**

The aim of this work was to elucidate how the dietary inclusion of phytase, at a normal dose and overdosed, could affect the utilization of nutrients and performance in young laying hens. When a diet deficient in Ca and P was applied, the dietary inclusion of phytase at low doses (500 FTU/kg) led to an improvement in the digestive efficiency of P in the first weeks after introduction. However, when these deficient diets were maintained in the long term, laying hens improved their digestive utilization of both Ca and P, a higher dose of phytase (1000 FTU/kg) being required to achieve greater P availability. This overdosage also provided additional extraphosphoric advantages, slightly improving access to other nutrients and the feed conversion rate of the hens.

**Abstract:**

A total of 192 laying hens were used to evaluate the effect of dietary mineral content and phytase dose on nutrient utilization, egg production and quality and bone mineralization of young laying hens. Four dietary treatments were studied: PC, positive control with no added phytase, 4.07% Ca and 0.61% P; NC, negative control with no added phytase, 2.97% Ca and 0.37% P; and P500 and P1000, where NC diet was supplemented with phytase at 500 and 1000 FTU/kg, respectively. Hens’ performance and egg traits were controlled from 22 to 31 weeks of age. Coefficients of total tract apparent digestibility (CTTAD) of nutrients were determined at 25 and 31 weeks of age. Apparent ileal digestibility (AID) and blood content of Ca and P, as well as bone traits, were determined at 31 weeks of age. Ca and P retention was higher in birds on PC diet at 25 weeks, but not at 31 weeks of age compared to those on NC diet (*p* < 0.05). P1000 birds had the highest CTTAD values for dry and organic matter at both ages (*p* < 0.001). CTTAD of Ca was significantly higher in P1000 diet than in NC diet at 31 weeks of age (*p* < 0.001). Birds fed with P500 diet at 25 weeks of age and P1000 at 31 weeks of age showed higher CTTAD and retention of P, but lower excretion of P than those fed NC diet (*p* < 0.05). Phytase inclusion linearly increased AID of dry matter and P (*p* < 0.001). P500 hens fed had the greatest body weight at the end of the trial (*p* < 0.05) and P1000 birds had the best feed conversion ratio (*p* < 0.05). Fowl fed a PC diet produced eggs with higher shell thickness and yolk color than those fed on NC diet (*p* < 0.05). Phytase inclusion linearly increased the yolk color (*p* < 0.05). Tibia of laying hens fed with PC had significantly higher ash content than those on NC diet (*p* < 0.05), and birds fed with P1000 presented intermediate values. It can be concluded that it would be advisable to increase the dose of phytase in the feed of laying hens to obtain long-term benefits.

## 1. Introduction

Phytases have been widely studied by scientists in the field of nutrition, environmental protection and biotechnology. These enzymes can release phosphate from phytate, which is the main form of phosphorus (P) storage in grains frequently used in animal diets. On the other hand, monogastric animals such as birds, pigs and fish lack phytase enzyme in their digestive system, or its activity is very low. Therefore, it is necessary to add mineral P in diets in the form of inorganic phosphate to meet the animals’ requirements. However, mineral phosphates are limited, expensive and non-renewable resources, and their use can cause environmental problems [1]. In addition, phytate is also known as an anti-nutritional agent, as phytate can form insoluble complexes, binding minerals and other nutrients (such as vitamins, proteins and amino acids), reducing their availability and absorption [2].

Production of high-quality eggs worldwide has a great impact on the economic dynamism of the egg industry. Laying hens have a metabolism that is highly dependent on the availability of minerals, such as Ca and P, in order to maintain effective egg production without compromising their mineral health status. This is especially important in young laying hens, where a reduction in the dietary level of minerals is promoted to encourage their feed intake [3] at peak laying time. Ca and P requirements seem to be affected by the hens’ age and production level, e.g., several studies have shown that the quality of eggshells decreases with age [4]. Additionally, the dietary Ca to P ratio is also relevant at these ages in laying hens [4,5]. Low rates reduce Ca absorption in the intestine, lead to decrease in shell quality, and could have severe long-term negative effects on Ca metabolism and bone reserves. However, high rates do not provide sufficient P and may cause a decrease of skeletal mineral content in laying birds. In this context, dietary inclusion of exogenous phytases could contribute to reducing mineral supplementation in young laying hens, promoting their feed intake, egg production and bone mineralization [6], although some studies have observed no positive effects [7].

Dietary exogenous phytase supplementation is widely used to improve dietary P availability in broilers [8], as phytase hydrolyzes the phytate present in grains, releasing the phytate-P, with the associated reduction in P excretion. However, the number of studies on the use of exogenous phytases in laying hens is smaller, and some authors affirm that the benefits of supplementing layer diets with phytase are still under discussion [5,9]. Although some authors have indicated that phytase inclusion in the diet at 250–500 FTU units can improve dietary P absorption [10,11], there is no consensus on its possible effect on improving dietary energy and protein utilization, and therefore, on laying hens’ performance and bone mineralization [8]. In fact, to improve the utilization of these nutrients, some authors mention that superdosing these exogenous phytases (1000 FTU or more) could eliminate phytates from the diet, contributing to an improvement in the nutritional value of the diet [12]. Another aspect not frequently considered is the possible appearance of compensatory effects when P-deficient diets are used, as some studies indicate that the length of trials could affect the efficacy of phytases in layers [5].

In this context, the present work is focused on elucidating how the inclusion level of phytases in P-deficient diets provided in short and long term could affect the utilization of P, but also the rest of the nutrients, and their possible effect on young laying hens’ performance. Therefore, the aim of this study was to evaluate the effect of dietary inclusion level of a 3-phytase at 500 and 1000 FTU/kg on nutrient digestibility and egg production and quality, as well as on bone mineralization in young laying hens from 22 to 31 weeks of age.

## 2. Materials and Methods

### 2.1. Animals and Housing

A total of 288 laying hens (Lohmann Brown) at 16 weeks of age were initially used. Animals came from a commercial breeding farm. The study lasted for 46 days of rearing and 60 days of experimental period. Upon arrival, hens were randomly distributed into 72 cages (4 animals/cage) located in two environmentally controlled rooms (36 cages per room). Cages (60 × 50 × 120 cm) were equipped with a feeder trough, two nipple drinkers and all the environmental enrichment elements according to Directive 1999/74/CE. At week 22 of age (day 1 of the experimental period), 192 hens were weighed and randomly distributed in 4 different treatments, with 12 replicates/treatment (48 animals/treatment). Experimental feeds were provided from week 22 of age until week 31 of age. At week 28 of age, an indigestible marker (titanium dioxide, TiO_2_) was added to the experimental diets at 4 g/kg. The marked feeds were provided until the end of the trial (week 31 of age). During the 106 days of study (rearing and experimental period), room temperature was controlled and maintained around 20 °C.

### 2.2. Experimental Diets

During the rearing period, all animals were fed a common commercial breeding feed based on corn, wheat and barley (weeks 16 and 17 of age) and laying feed based on wheat and corn (weeks 18 and 21 of age) until they reached a daily production of 1.07 eggs per hen. At 22 weeks of age, hens were assigned to one of the four dietary treatments: PC, positive control with no added phytase: Ca at 4.07% and P at 0.61%; NC, negative control with no added phytase: Ca at 2.97% and P at 0.37%; and two other diets in which NC diet was supplemented with ePhyt 1000^®^ 3-phytase (Globalfeed) at 500 (P500) and 1000 (P1000) FTU/kg feed, respectively (see enzyme details at Cambra-López et al. [13]). All the diets were formulated following the recommendations given for laying hens by FEDNA [14], except for Ca and P in the negative control, being isonutritive for the rest of the nutrients (Table 1). Feed and water were provided ad libitum throughout the experiment and diets were fed in mash form.

### 2.3. Laying Performance and Egg Quality

Body weight (BW) was recorded per cage on arrival (16 weeks of age), at the start of the administration of experimental feeds (22 weeks of age) and at weeks 25 and 31 of age of the experimental period. Feed consumption was recorded at each weighing control to calculate average daily feed intake (DFI). Health status of the animals was checked daily and necropsies were performed from all dead animals. The number of eggs laid and their weight were daily monitored. Average laying index (egg/hen and day), egg weight (g) and egg mass (laying index x egg weight; g/day) were determined weekly. Average daily feed intake (DFI; g/day) and feed conversion ratio (FCR; g feed/g egg) were also calculated globally. Furthermore, the number of eggs with shell quality problems (soft shelled eggs, shell-less eggs) was registered daily. On days 52, 53 and 59 of the experimental period, all the eggs laid in the last 24 h (approximately 150 eggs/treatment) were collected for egg quality measurements. The sampled eggs were individually weighed and broken on a flat surface. Subsequently, the height of the inner thick albumen (Haugh units) was measured with an electronic albumen height gauge. The Haugh units were calculated 100 × log_10_ (H + 7.57 − 1.7W^0.37^), where H is the height of the albumen and W is the weight of the egg, according to Haugh [15]. The shells were broken in three parts and shell thickness was a mean value of measurements at these three locations taken by using a dial pipe gauge (3001 digital Baxlo, Instrumentos de Medida y Precisión S.L., Barcelona, Spain). Additionally, yolk color was determined by the Roche yolk color fan (Hoffmann-La Roche Ltd., Basel, Switzerland; color scale from 15, dark orange, to 1, light pale).

### 2.4. Fecal and Ileal Digestibility

At weeks 25 and 31 of age, a nutrient retention balance was performed. Total excreta output and feed intake were measured quantitatively per cage (12 cages/treatment) for two days. During each of these 2-day collection periods, excreta were collected every 24 h, weighed and stored at 4 °C. At the end of the collection period, excreta were pooled per cage and homogenized. Representative samples were then taken and stored at −20 °C until analysis. Feed samples were dried at 105 °C for 24 h and then ground up. Excreta samples were dried at 80 °C for 48 h and then ground. Dry matter (DM), ash, crude protein (CP), gross energy (GE), Ca and P were determined in feeds and excreta samples. Ether extract (EE) and phytate-P and TiO_2_ were also determined in feeds.

The coefficients of total tract apparent digestibility (CTTAD) for DM, organic matter (OM), GE, CP, Ca and P were calculated using the following equation:$$CTTAD\;\left(\%\right)=\frac{\left[\left(Feed\;intake\times Nutrient_{feed}\right)-\left(Excreta\;output\times Nutrient_{excreta}\right)\right]}{\left(Feed\;intake\times Nutrient_{feed}\right)}\times 100$$

At the end of the trial (31 weeks of age), all birds were euthanized by stunning and exsanguination to obtain the ileal content. The ileum was removed by cutting the portion of the small intestine from Meckel’s diverticulum to about 5 mm proximal to the ileocecal junction [11,16]. A 4 mL syringe full of room temperature distilled water was inserted at one end of the ileum and the digesta were carefully flushed out of the gut into a 10 cm diameter Petri dish [17,18]. The digesta from all birds in a cage were pooled and stored at −80 °C until laboratory analysis. Ileal content was lyophilized and analyzed for DM, TiO_2_, Ca and P.

The apparent ileal digestibility (AID) of Ca and total P was calculated by the relation:$$AID\;\left(\%\right)=\left[1-\left(\frac{TiO_{2\;feed}\times Mineral_{digesta}}{TiO_{2\;digesta}\times Mineral_{feed}}\right)\right]\times 100$$

### 2.5. Bone Mineralization and Blood Analysis

From the animals slaughtered at the end of the trial, one bird per cage (12 animals per treatment) was randomly selected to evaluate bone mineralization. The left tibia from this animal was obtained and frozen, after removing all the soft tissues, at −20 °C until analysis. Tibias were boiled to remove the remaining soft tissues, cleaned and dried at 110 °C for 12 h. Then, tibias were degreased in an ether solution for 48 h. Once cleaned and degreased, tibias were dried again at 110 °C for 12 h, weighed and then ash, Ca and P content was determined.

Blood samples from each animal were also collected at 31 weeks of age, into two 4 mL vacutainer tubes with serum clot activator, refrigerated and transported to the laboratory to determine Ca and P content.

### 2.6. Analytical Methods

DM (934.01), ash (942.05), EE (920.39) with acid hydrolysis prior to ether extraction and CP (990.03) were analyzed according to AOAC methods [19]. GE was determined using an adiabatic bomb calorimeter (Gallenkamp Autobomb, Loughborough, UK). Mineral (Ca and P) content was analyzed by inductively coupled plasma atomic emission spectrometry (ICP-OES) (model Varian 720-ES, Varian Inc., Palo Alto, CA, USA), as described in Cambra-López et al. [19]. Phytate-P was analyzed by spectrophotometry according to the method described by Haug and Lantzch [20]. TiO_2_ concentration was analyzed in feeds and ileal content according to the methodology proposed by Short et al. [21].

### 2.7. Statistical Analyses

Data were analyzed using SAS System Software (version 9.1, SAS Institute Inc., Cary, NC, USA). The experimental unit was the cage for ADFI, FCR, body weight, egg production and nutrient balance traits; the egg for the egg quality traits; and the hen for the mineral content in tibia and blood.

Data on hen performance and egg production traits were analyzed in a repeated measures design taking into account the variation between animals and covariation within them. Covariance structures were objectively compared using the strictest criteria (Bayesian information criterion; [22]). The model included the treatment (PC, NC, P500 and P1000), the age (25 and 31 weeks of age) and their interactions as fixed effects. Random terms in the model included a permanent effect of each animal (p) and the error term (e), both assumed to have an average of zero, and variance $\sigma_{p}^{2}$ and $\sigma_{e}^{2}$. Data on CTTAD, AID, egg quality, bone and blood traits were analyzed according to the general lineal model (GLM) in a completely randomized design with a model accounting for the fixed effect of the treatment (PC, NC, P500 and P1000), the age (25 and 31 weeks of age) and their interactions. Additionally, polynomial orthogonal contrasts were applied to test linear (L) effects among treatments NC, P500 and P1000. Results were presented as least square means with their standard error of the means (SEM). Statistical significance level was set at 5% (0.05).

## 3. Results

Table 2 shows the effect of dietary phytase inclusion on nutrient CTTAD, as well as on mineral retention and excretion for the laying hens’ diets at 25 and 31 weeks of age. There was a clear effect of diet (mineral level and phytase) on all these parameters, as well as a significant diet x age interaction for CP, GE and P CTTAD; Ca and P retention; and P excretion. In general, the CTTAD of main nutrients was significantly higher at 31 than 25 weeks of age (*p* < 0.001). Among diets, animals fed with the PC diet showed higher CTTAD of OM, CP and GE, as well as Ca retention and Ca and P excretion, but lower CTTAD of Ca and P, compared to those fed on NC diet (*p* < 0.05). Phosphorus retention was higher in animals on the PC diet at 25 weeks of age, but not at 31 weeks of age (diet x age; *p* < 0.05).

Regarding the effect of phytase on nutrient digestibility, inclusion of the 3-phytase diet increased CTTAD of DM and OM at 25 and 31 weeks of age. Animals fed with P1000 had the highest CTTAD values for DM and OM at both ages (+1.0 and +1.7 percentage points compared to NC, respectively; *p* < 0.001). The effect of phytase inclusion on CTTAD of CP and GE was different depending on the age (*p* < 0.01). The CTTAD of CP in P1000 diet was higher at 25 weeks of age (*p* = 0.02) but lower at 31 weeks of age than in NC diet (*p* < 0.002). Regarding CTTAD of GE, it was significantly lower in P500 than in NC diet at 31 weeks of age (*p* < 0.05). As for mineral utilization, the CTTAD of Ca was significantly higher in P1000 diet than in NC diet at 31 weeks of age (*p* < 0.001). Finally, dietary phytase inclusion improved CTTAD, retention and excretion of P at both ages, although at different levels of inclusion depending on the age (diet x age; *p* < 0.05). Animals fed with P500 diet at 25 weeks of age and P1000 at 31 weeks of age showed higher CTTAD (*p* < 0.05) and retention of P (*p* < 0.05), but lower excretion of P (*p* < 0.05) than those fed with NC diet.

Apparent ileal digestibility (AID) of Ca and P in 31-week-old layers, as well as their concentration in blood, is presented in Table 3. In general, values for AID of DM, Ca and P were lower than those of CTTAD presented in Table 2. Animals fed with PC diet showed lower Ca ileal digestibility and higher Ca and P concentration in blood than those on NC diet (*p* < 0.05). Dietary inclusion of the 3-phytase did not affect Ca ileal digestibility or blood concentration of Ca and P at 31 weeks of age. However, AID of both DM and P with P1000 diet was significantly higher than with NC diet (*p* < 0.001).

Table 4 presents the effect of dietary treatments on hens’ performance and egg production traits from 22 to 31 weeks of age. Figure 1 shows that weekly egg mass evolution throughout the study was not affected at any time by the different dietary treatments. Although hens’ performance was not significantly affected by mineral level of the diet, animals fed with PC diet produced eggs with a higher shell thickness and yolk color than those on NC diet (*p* < 0.05). Regarding the effect of phytase inclusion, hens fed with P500 diets had the greatest body weight at the end of the trial (*p* < 0.05) and those on P1000 diets had the best FCR (*p* < 0.05). Hens fed with P1000 diets also had the lowest shell thickness values (*p* < 0.05). Dietary inclusion of phytase linearly increased the yolk color (*p* < 0.05), allowing us to achieve the values reached with the PC diet.

Finally, Table 5 shows the effect of diets on bone mineralization of young laying hens after 9 weeks of the treatment. Mineral level of the diet significantly affected main tibia mineralization traits. Tibia of laying hens fed with PC had significantly higher ash, Ca and P content than those with NC diet (*p* < 0.05). Dietary phytase inclusion did not significantly affect the main mineralization traits controlled. However, tibia ash content of animals fed with P1000 had intermediate values, which were not significantly different from those of hens on PC diet.

## 4. Discussion

### 4.1. Nutrient Utilization

Domestic animals excrete about 15 million tons of phosphorus into the environment every year [23]. Numerous studies have reported that mineral supplementation in commercial feeds increases phosphorus intake and its excretion rate, which could cause serious environmental problems, as phosphorus sources are limited and non-renewable resources [1,24,25,26]. In our study, mineral supplementation below commercial levels led to higher CTTAD and AID of Ca and higher CTTAD of P, regardless of the age of the young laying hens. In fact, higher mineral provision with PC diet (+25% Ca and +64% P), together with lower mineral digestibility compared to NC diet, led to higher Ca and P excretion. Other studies also showed higher Ca and P digestibility when diets were deficient in these minerals, indicating that this lower provision in minerals might increase their digestive efficiency. For instance, some studies have observed an increase in P digestibility in hens fed with P-deficient diets, both at fecal and ileal level [27], reducing P excreta [28]. In fact, Ren et al. [29] reported that when dietary inorganic P was overdosed, it was mainly excreted by the laying hens. Regarding Ca, although some studies observed a decrease in Ca digestibility when dietary inorganic P was overdosed [29], others have observed the opposite behavior, decreasing Ca digestibility in Ca-deficient diets [27,30].

The effect of a diet deficient in Ca and P on the use and retention of these minerals seems to change as we maintain this deficit over time. As expected, after only 3 weeks on the experimental diets, the hens with the deficient diet showed a lower daily retention of Ca and P, with a slight reduction in their excretion at 25 weeks. However, after 9 weeks on the deficient diet, these animals seemed to improve their digestive utilization of Ca and P, with no differences being observed in the daily retention of Ca, and this improvement was even greater for those on dietary P compared to those fed with PC, significantly reducing the excretion of both minerals. Recently, Bello and Korver [31], in laying hens receiving nutritionally adequate and deficient diets in P from 30 to 70 weeks of age, also observed that hens fed the deficient diet increased both AID of P (from 40 to 53%) and the P retained (from 0.20 to 0.25 g/d) from 32 to 48 weeks of age. This is a relevant issue, as it could also affect the evaluation of phytase effectiveness in the long term.

Despite this improvement in daily mineral retention, the levels of Ca and P in the blood remained lower with the deficient diet. Previous works, where dietary Ca and P levels were similar to those evaluated in this research, showed contradictory results. Some studies observed that an increase in the dietary level of Ca and P did not lead to relevant modifications of these minerals in the blood [29,32,33]. However, Sari et al. [27] reported that a decrease in the P level of the diet led to a clear decrease in the blood P level of the hens (8.01 vs. 4.10 mg/dL), but without modifying the serum Ca level. In fact, Viveros et al. [34] described a linear correlation between dietary non-phytate P and plasma P for different poultry species. Ren et al. [29] associated these differences with the blood sample-collecting time used in the different trials.

Regarding the effects of dietary addition of phytase on mineral digestibility, retention and excretion, in the present work a clear interaction between the phytase level and age was found for P utilization. At 25 weeks of age, the P digestibility and retention were higher (*p* < 0.05) and P excretion was lower with the diet including phytase at 500 FTU/kg compared to the NC diet. However, at 31 weeks of age, the highest P digestibility (both fecal and ileal) and retention and the lowest P excretion were found in the group of animals fed the diet including 1000 FTU/kg. Therefore, the results of the present work could indicate that the recommended dose of the phytase for an effective use of P could be age-dependent, with a higher dose being required as the age of laying hens increases. The inclusion of phytase in laying hen diets can increase P digestibility and retention, as phytase hydrolyzes the phytate present in grains, releasing the phytate-P. However, the effective dose of phytase might change depending on the phytase, diet, other mineral levels and age, among other factors [5,35]. Although there are not many studies evaluating the effect of age on the effectiveness of phytases in laying hens, most authors agree that their effectiveness decreases with age. Van der Klis et al. [36] found that Ca and P absorption at 36 weeks of age was significantly lower than at 24 weeks. More recently, a meta-analysis of the studies carried out with phytases in laying hens [5] described a negative correlation between age and the efficacy of phytase in terms of retention of P. These results could indicate that when diets with low levels of non-phytic P are used, the level of inclusion of phytase should increase with the age of the hens to ensure an adequate supply of P to these animals. However, we must also take into account in these studies the time from the introduction of mineral-deficient diets. As we have noted in the present work, laying hens can increase their Ca and P absorption efficiency when they are receiving a deficient diet for a long time, through an increase in the metabolism of renal and intestinal 1,25-hydroxycholecalciferol [37]. In this sense, Bello and Korver [31] observed how phytase supplementation in deficient diets at 30 weeks of age improved AID of P at 32 weeks of age, but this advantage disappeared thereafter (at 48 and 70 weeks of age). For this reason, we should avoid short-term trials to evaluate phytase effectiveness, as their commercial use will be long term in deficient diets, and we should probably recommend higher doses than those applied in short-term trials.

The highest values for dietary Ca utilization (both at fecal and ileal level) were obtained when P was also better used, at 31 weeks of age when phytase was overdosed at 1000 FTU/kg. Sometimes, the improvement observed in Ca digestibility with phytase inclusion is the consequence of a drop in digestibility in the control P-deficient diets, as there is an increase in the Ca/P ratio that can promote the formation of insoluble Ca phosphate and reduce Ca solubility in the digestive tract [30]. In this work, there was no such fall of Ca digestibility in the NC diet compared to the PC diet, as the Ca/P ratio was barely modified. Therefore, we can assume that the improvement in the Ca utilization observed was mainly due to the fact that phytases are able to liberate not only P, but also Ca from Ca-phytate complexes [8].

Finally, when we overdosed the phytase at 1000 FTU/kg, an improvement in the use of DM and OM, even in that of CP at 25 weeks of age, was observed. In broilers, Dersjant-Li and Kwakernaak [38] reported a linear increase in both ileal digestibility of total amino acids and apparent metabolizable energy (AME), its effect being different depending on the phytase used and independently of the available P. Selle et al. [8], reviewing the main mechanisms proposed by the literature for the extraphosphoric effects, proposed that phytate could reduce the digestive utilization of dietary protein and energy by binding to amino acids, increasing mucin and then the loss of endogenous protein and compromising the Na+-dependent transport of starch, glucose and amino acids in the gut. In fact, Lei et al. [39] observed that the CP and AME content of laying hens’ diets could be slightly reduced thanks to the extraphosphoric consequences of phytase supplementation without penalties. However, these benefits could be slightly reduced in the long term, and this should be considered when formulating diets.

### 4.2. Laying Hens’ Performance and Egg Quality

Differences in the Ca and P levels between PC and NC diets did not affect laying hens’ performance during the 60-day experimental period, thus suggesting that laying hens can maintain optimal medium-term performance when fed a diet containing 2.0 g/kg non-phytate P (nPP), if feed intake is maintained within normal values. Previous reports indicated that diets containing 2.0–2.3 g/kg available P (aP) are enough to maintain hen performance when dietary Ca is within the range of 32.5–40.0 g/kg [40,41,42,43]. Boling et al. [10] reported that P deficiency signs in older hens (70 weeks) occurred within only 3 weeks of consuming a diet with 1.0 g/kg aP, compared to 8 weeks in younger hens (20 weeks). The authors suggested that older hens may exhibit P deficiency symptoms sooner than younger hens. However, it seems that there are dietary interactions between Ca and P in high egg-producing layers, as significant performance depression and high mortality rates are seen when low P content is combined with high Ca in the diet [44]. As the Ca/P ratio was not excessively modified in the present work, these young laying hens were not expected to show alterations in their reproductive performance when fed with a deficient diet from 22 to 31 weeks of age.

In terms of phytase addition, the dietary inclusion of the 3-phytase in the present study increased hens’ final weight at 500 FTU/kg inclusion and improved FCR at 1000 FTU/kg. As mentioned above, extraphosphoric effects of phytase inclusion allow greater availability of other nutrients, especially when phytase is overdosed, which could slightly contribute to improving laying hens’ performance. Similar results have previously been reported in other works [34,45], supporting the idea that the inclusion of phytases could allow a slight reduction in the level of other nutrients in the diet [39].

Literature results indicate that diets with 0.15–0.25% nPP and in the absence of phytase [36,46,47] and diets with 0.10–0.15% nPP supplemented with phytase [46,47,48,49] are sufficient to maintain satisfactory egg production performance during the laying cycle. Hughes et al. [7] showed no significant differences in egg production traits of laying hens fed with diets either containing 3.5 or 2.5 g/kg nPP, but those fed with 1.5 g/kg nPP had significantly reduced egg performance and higher incidence of soft-shelled and broken eggs compared to 3.5 g/kg nPP. In fact, the literature suggests that the addition of phytase to 1.0 and 2.0 g/kg nPP diets for hens could improve the hens’ weight and feed efficiency (feed to egg mass ratio; [50,51]), but in general it seems that extremely low levels of P are needed to affect these parameters [35,52].

In our study, where diets with 2.0 and 4.4 g/kg nPP were compared, we did not observe significant differences in egg production, with or without phytase addition, but we observed both greater shell thickness and value for the yolk color with the diet including 4.4 g/kg nPP. Laying hens require Ca to form amorphous calcium carbonate and calcium phosphate during eggshell calcification [53,54]. However, most of the works reviewed in the literature show that the main determinant of the quality of the shell is the level of Ca and not so much the level of P. Bar et al. [55] already observed that an increase in the Ca level produced a clear improvement in the shell weight, while the modifications of the P level had no effect. In fact, most of the studies that evaluated the effect of the inorganic P level, at a constant Ca level, did not observe any significant effect on the eggshell characteristics [29,32,36,56]. These results could explain why the PC diet allowed obtaining eggs with a greater shell thickness compared to NC diet, by providing a higher level of Ca, while the greater availability of P due to the inclusion of phytase did not lead to improvements in the shell quality. Regarding the yolk color, there seems to be an association between the dietary level of P and the intensity of yolk color. Several authors [43,57,58] have observed an increase in the intensity of the yolk color when they increased the level of inorganic P in the feed. In addition, several studies have reported a similar effect when phytase is added to feed [38,43,59,60]. Brunelli et al. [59] associated this effect with the hydrolysis of phytic acid, as phytic acid has depigmenting properties [61].

### 4.3. Bone Mineralization

Tibia quality has long been used to evaluate the phosphorus requirement of poultry species because it is a more sensitive indicator of phosphorus sufficiency than productive performance. In the present study, animals fed the NC diet showed lower ash, Ca and P retention in tibia compared with animals fed the PC diet, indicating that hens with the deficient diet started mobilizing bone mineral to support their eggshell formation. Similar results have also been observed in other short-term trials. Pongmanee et al. [30] reported that laying hens fed with a Ca- and P-deficient diet from 25 to 37 weeks of age had significantly lower bone mineral density and content when compared with a diet meeting hens’ requirements.

As regards phytase, previous studies showed that dietary phytase inclusion could increase bone Ca and P concentrations, breaking strength and ash content in laying hens fed Ca- and P-deficient diets after 17–22-week trials or in old laying hens [1,42]. The percentage of P in tibia was not affected by phytase inclusion after 9 weeks of trial in this work, but a dose of 1000 FTU/kg in diets slightly increased Ca and ash content in tibia, reaching the levels found in the animals fed the PC diet. It seems that positive effects of phytase addition are more pronounced in older laying hens and long-term trials. Hughes et al. [7] found that phytase addition to a deficient diet did not affect bone ash percentage at 42 weeks of age, but it was significantly improved at 61 weeks of age. In any case, there are already several studies indicating that, when enough phytase is introduced in the feed (2000 FTU/kg), the level of aP is not a limiting factor for the bone structure of laying hens in the long term [29,56].

## 5. Conclusions

The results of this work allow us to conclude that when a diet deficient in Ca and P was applied, the dietary inclusion of phytase at low doses (500 FTU/kg) afforded an improvement in the digestive efficiency of P during the first weeks after introduction. However, when this type of deficient diet is maintained in the long term, laying hens seem to improve their capacity for digestive utilization of both Ca and P, and it is necessary to include a higher dose of phytase (1000 FTU/kg) to achieve greater availability of dietary P. On the other hand, this overdosage allowed a series of additional extraphosphoric advantages, slightly improving access to other nutrients and the feed conversion rate of the hens, as well as favoring the recovery of some traits related to shell quality and bone mineralization that worsened with the deficient diet. Therefore, due to these compensation phenomena and the possible extraphosphoric effects, it would be advisable to increase the dose of phytase in the feed for laying hens in order to achieve long-term benefits.

## Figures and Tables

**Figure 1 animals-11-01495-f001:**
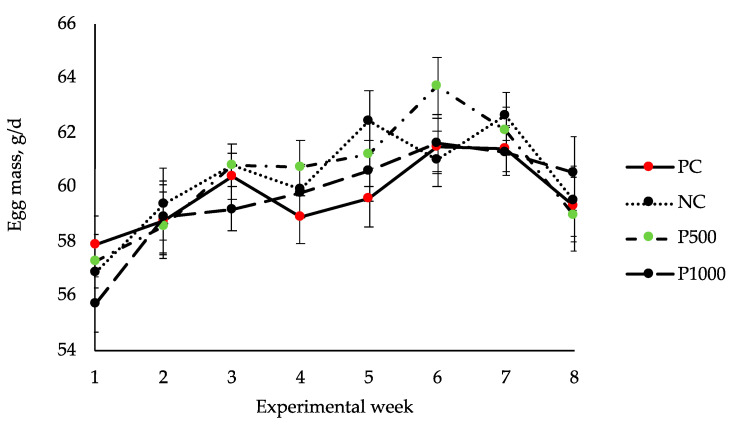
Egg mass evolution over the trial. Treatments: PC, positive control; NC, negative control; P500, NC with 500 FTU/kg of phytase and P1000, NC with 1000 FTU/kg of phytase. *p*-value of treatment = 0.614, *p*-value of week < 0.001, *p*-value of treatment × week = 0.724. Bars represent standard errors.

**Table 1 animals-11-01495-t001:** Ingredients and chemical composition of the experimental diets (g/kg).

Ingredients and Chemical Composition	Positive Control	Negative Control
Ingredients		
Corn grain	592.5	610
Soybean meal 44% CP	276	276
Soybean meal oil	25.4	25.4
DL-Methionine	1.6	1.6
Calcium carbonate	80.0	75.0
Dicalcium phosphate	16.0	3.5
Salt	3.45	3.45
Red coloring (canthaxanthin 10%)	0.05	0.05
Vit-min premix ^1^	5	5
Chemical composition		
Dry matter	899	900
Ash	118	111
Crude protein	170	172
Ether extract	44.1	44.4
Gross energy (kcal/kg)	3652	3815
Apparent metabolizable energy (kcal/kg)	2819	2913
Metabolizable protein	90	87
Calcium	40.7	29.7
Total phosphorus	6.1	3.7
Phytate phosphorus	1.7	1.7
Non-phytate phosphorus ^2^	4.4	2.0

^1^ Provides per kilogram of feed: calcium: 200.61 g, E5 manganese (manganese oxide): 13,000 mg, E6 zinc (zinc oxide): 7400 mg, E4 copper (copper sulphate pentahydrate): 800 mg, E2 iodine (potassium iodide): 380 mg, E8 selenium (sodium selenite): 20 mg, E1 iron (carbonate ferrous): 3600 mg, E672 vitamin A: 1,500,000 UI, E671 vitamin D3: 300,000 UI, vitamin K: 300 mg, vitamin B2: 600 mg, vitamin B12: 2000 mg, niacin: 3000 mg, calcium pantothenate: 1400 mg, pantothenic acid: 1288 mg, betaine: 10,830 mg, choline chloride: 25,500 mg, E320 butylhydroxyanisol (BHA): 4 mg, E321 butylhidroxytoluene (BHT): 44 mg, E324 ethoxyquin: 6.40 mg, dry matter: 956.54 g. ^2.^ Calculated as the difference between total phosphorus and phytate phosphorus.

**Table 2 animals-11-01495-t002:** Effect of mineral level (diet) and phytase inclusion on nutrient coefficient of total tract apparent digestibility (CTTAD, %), retention (ret; g/d animal) and excretion (exc; %) of diets in laying hens at 25 and 31 weeks of age.

					Ca			P		
	DM	OM	CP	GE	CTTAD	Ca Ret	Ca Exc	CTTAD	P Ret	P Exc
25 weeks of age										
PC	71.3 ^b^	74.9 ^a^	50.4 ^b^	76.8 ^a^	57.3 ^b^	2.54 ^a^	42.7 ^a^	20.8 ^b^	0.134 ^a^	79.2 ^a^
NC	71.8 ^b^	73.4 ^b^	46.0 ^c^	75.9 ^b^	63.6 ^a^	2.05 ^b^	36.4 ^b^	24.2 ^b^	0.089 ^b^	75.8 ^a^
P500	72.3 ^ab^	74.7 ^a^	51.9 ^ab^	76.7 ^ab^	63.5 ^a^	2.44 ^a^	36.5 ^b^	30.1 ^a^	0.128 ^a^	69.9 ^b^
P1000	72.8 ^a^	75.1 ^a^	53.0 ^a^	76.7 ^ab^	65.2 ^a^	2.05 ^b^	34.8 ^b^	24.9 ^b^	0.101 ^b^	75.1 ^a^
SEM	0.36	0.331	0.761	0.341	1.52	0.082	1.52	1.65	0.011	1.65
31 weeks of age										
PC	71.8 ^b^	75.9 ^a^	55.2 ^a^	77.6 ^a^	58.3 ^c^	2.68 ^a^	41.7 ^a^	10.1 ^c^	0.058 ^c^	90.9 ^a^
NC	72.7 ^b^	74.6 ^b^	55.0 ^a^	76.8 ^ab^	67.3 ^b^	2.62 ^ab^	32.7 ^bc^	31.7 ^b^	0.129 ^b^	68.3 ^b^
P500	72.2 ^b^	74.6 ^b^	54.1 ^ab^	75.3 ^c^	64.6 ^b^	2.50 ^ab^	35.4 ^b^	29.9 ^b^	0.120 ^b^	70.1 ^b^
P1000	73.9 ^a^	76.2 ^a^	52.2 ^b^	76.5 ^b^	70.6 ^a^	2.47 ^b^	29.4 ^c^	38.9 ^a^	0.179 ^a^	61.1 ^c^
SEM	0.445	0.348	0.994	0.389	1.66	0.086	1.66	2.15	0.011	2.19
*p*-value										
Diet	<0.001	<0.001	0.02	0.002	<0.001	<0.001	<0.001	<0.001	<0.001	<0.001
Age	0.014	<0.001	<0.001	0.856	0.008	<0.001	0.008	0.052	0.208	0.077
Diet x Age	0.344	0.117	<0.001	0.002	0.366	0.006	0.366	<0.001	<0.001	<0.001

^a,b,c^ Means within a column and age not sharing superscripts differ at *p* < 0.05. Treatments: PC, positive control; NC, negative control; P500, negative control with phytase at 500 FTU/kg feed and P1000, negative control with phytase at 1000 FTU/kg feed. SEM: standard error of the mean. DM: dry matter; OM: organic matter; CP: crude protein; GE: gross energy; Ca ret: Ca retained; Ca exc: Ca excreted; P ret: P retained; P exc: P excreted.

**Table 3 animals-11-01495-t003:** Apparent ileal digestibility (AID) and blood concentration of calcium (Ca) and phosphorus (P) of laying hens at 31 weeks of age.

Traits	PC	NC	P500	P1000	SEM	*p*-Value
AID, %						
Dry matter ^1,2^	68.24 ^b^	68.82 ^b^	66.58 ^b^	76.50 ^a^	1.37	<0.001
Ca	42.70 ^c^	59.89 ^ab^	53.14 ^bc^	65.60 ^a^	4.27	0.003
P ^1^	19.89 ^b^	22.91 ^b^	29.67 ^b^	52.96 ^a^	3.52	<0.001
Blood concentration, mg/dL						
Ca	30.47 ^a^	28.35 ^b^	28.01 ^b^	27.87 ^b^	0.721	0.043
P	7.35 ^a^	5.38 ^b^	5.56 ^b^	5.98 ^b^	0.368	0.002

^a,b,c^ Means with different superscripts differ (*p* < 0.05). Treatments: PC, positive control; NC, negative control; P500, negative control with 500 FTU/kg of phytase and P1000, negative control with 1000 FTU/kg of phytase. SEM: standard error of the mean. ^1^ Linear effect of the phytase inclusion. ^2^ Quadratic effect of the phytase inclusion (*p* < 0.05).

**Table 4 animals-11-01495-t004:** Effect of dietary phytase inclusion on performance and egg production traits of laying hens from 22 to 31 weeks of age.

Traits	PC	NC	P500	P1000	SEM	*p*-Value
Initial body weight, g	1749	1779	1753	1747	0.024	0.759
Final body weight, g	1872 ^ab^	1851 ^b^	1901 ^a^	1840 ^b^	0.017	0.074
ADFI, g/day	109.2	108	110.5	105.2	1.77	0.176
FCR, g feed/g egg	1.849 ^a^	1.799 ^ab^	1.830 ^ab^	1.758 ^b^	0.031	0.187
Average laying index	0.973	0.975	0.978	0.98	0.044	0.528
Average egg mass, g/day	59.28	60.56	59.66	59.84	0.696	0.594
Egg traits at 31 weeks of age:						
Shell thickness, mm ^1,2^	0.382 ^a^	0.371 ^b^	0.375 ^ab^	0.359 ^c^	0.003	<0.001
Albumen height, mm	11.46	11.24	11.44	11.37	0.152	0.725
Haugh units	104.3	103.4	104.2	104.2	0.595	0.696
Yolk color ^1,3^	13.81 ^a^	13.57 ^b^	13.80 ^a^	13.95 ^a^	0.059	<0.001

^a,b,c^ Least square means in a row not sharing superscripts differ at *p* < 0.05. Treatments: PC, positive control; NC, negative control; P500, negative control with 500 FTU/kg of phytase and P1000, negative control with 1000 FTU/kg of phytase. SEM: standard error of the mean; ADFI: average daily feed intake; FCR: feed conversion ratio. ^1^ Linear effect of the phytase inclusion (*p* < 0.05). ^2^ Quadratic effect of the phytase inclusion (*p* < 0.05). ^3^ Points in the Roche scale.

**Table 5 animals-11-01495-t005:** Effect of dietary phytase inclusion on bone mineralization traits of laying hens at 31 weeks of age.

Traits	PC	NC	P500	P1000	SEM	*p*-Value
Tibia weight, g	6	5.96	5.81	5.81	0.197	0.845
Tibia weight, % BW	0.322 ^a^	0.308 ^ab^	0.301 ^b^	0.299 ^b^	0.007	0.113
Ash in tibia (% DM)	52.0 ^a^	49.8 ^b^	49.6 ^b^	50.8 ^ab^	0.463	0.001
Ca in tibia (% DM)	18.72 ^a^	18.09 ^b^	17.92 ^b^	18.29 ^ab^	0.235	0.064
P in tibia (% DM)	8.78 ^a^	8.39 ^b^	8.38 ^b^	8.51 ^b^	0.085	0.001

^a,b^ Least square means in a row not sharing superscripts differ at *p* < 0.05. Treatments: PC, positive control; NC, negative control; P500, negative control with 500 FTU/kg of phytase and P1000, negative control with 1000 FTU/kg of phytase. DM: dry matter; BW: body weight. SEM: standard error of the mean.

## Data Availability

Data are contained within the article.

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
