# Peer review of "Effect of Dietary Mineral Content and Phytase Dose on Nutrient Utilization, Performance, Egg Traits and Bone Mineralization in Laying Hens from 22 to 31 Weeks of Age"

_animals, 2021, doi:10.3390/ani11061495_

Round 1

Reviewer 1 Report

It is generally well written work concerning the actual problem of poultry production. I find the article interesting for the audience but some concerns need to be addressed before I can consider this manuscript acceptable for publication.

Simple summary: It is too technical. It should present the general idea of the study and results described in non-technical manner.

L47 Please rephase: phytases are studied for over 20 years, not “In recent years”

L49 consider changing “P storage in the plant seeds” to “P storage in grains

L56 Please remove “…”

L77 in this paragraph, you are referring to 15-20 y old studies. Please give more recent references which can support your statements

L85-90 Introduction does not explain in sufficient way why bones traits were analyzed. The information how bone quality (mineralization) influence the egg performance and eggshell quality is missing (we have just a single note in L69).

L100 What type of experimental design was used?  LSD, CRD, RBD or without blocking?

L103, L162 and others: Ti02 – subscript

L110 1.07 egg per hen per day? ?

L115 Please correct reference style

L120 Why eggshell weight/percentages were not recorded? It is as essential information regarding eggshell mineralization as shell thickness. What was the idea of measuring albumen thickness and HU? what is the connections of HU or albumen height with Ca, P or phytase supplementation? Moreover, results of these traits are not discussed at all.

L134 HU is not measured in %

L135 Please double-check HU equation

Table 1 – please mark which parameters were calculated and which were determined in analyses.

L172 Were bones crushed to make possible bone marrow removal during boiling/cleaning/degreasing? Cartilage caps were also removed after boiling ?

L197 Term BIC is more commonly used.

L208 Please consider removing reporting percentage points changes. You are refereeing to mean/lsmean values, and some traits were determined with an accuracy of percentages

L231 diet x age

L236-243 The description of AID results (Table 3) is insufficient. There is no information about quadric effect for DM. Please also report exact p-values for contrast analysis

Figure 1  Whishes show SD or SE?

Table 5 Some data presented in this table should be deleted. It is just multiple presentation of the same results. Only significant changes were observed when ash/Ca/P were expressed in relation to DM, which  is a proper presentation of the ash/mineral content in bones.

L317 “8.01 vs. 4.10” units?

L386 the information that "a new 3-phytase" was supplemented in this study is missing in introduction and materials and methods sections. Please provide same characteristic of this new phytase.

Author Response

Reviewer #1

It is generally well written work concerning the actual problem of poultry production. I find the article interesting for the audience but some concerns need to be addressed before I can consider this manuscript acceptable for publication.

Simple summary: It is too technical. It should present the general idea of the study and results described in non-technical manner.

Authors: We agree with the reviewer, we have completely rewritten the simple summary in a non-technical manner, providing the general idea of the study: “This work has been addressed at elucidating how the dietary inclusion of phytase, at a normal dose and overdosed, could affect the utilization of nutrients and young laying hens’ performance. When a diet deficient in Ca and P has been applied, the dietary inclusion of phytase at low doses (500 FTU/kg) allowed to obtain an improvement in the digestive efficiency of P during the first weeks after introduction. However, when these deficient diets are maintained in the long-term, laying hens improved their digestive utilization of both Ca and P, being needed a higher dose of phytase (1000 FTU/kg) to obtain greater P availability. This over-dosage also allowed additional extra-phosphoric advantages, slightly improving access to other nutrients and the feed conversion rate of the hens.”

L47 Please rephase: phytases are studied for over 20 years, not “In recent years”

Authors: Ok, “In recent years” has been deleted.

L49 consider changing “P storage in the plant seeds” to “P storage in grains

Authors: Done.

L56 Please remove “…”

Authors: Done.

L77 in this paragraph, you are referring to 15-20 y old studies. Please give more recent references which can support your statements

Authors: We have also included cite [5] in the reference that support this statement. This meta-study also highlights the heterogeneity in the effects of phytases in layers related to experiment length, animals age, Ca content and phytase dose.

L85-90 Introduction does not explain in sufficient way why bones traits were analyzed. The information how bone quality (mineralization) influence the egg performance and eggshell quality is missing (we have just a single note in L69).

Authors: Ok. We considered that it was explicit in the term "performance", but we have included it in the following sentence to justify its analysis: “…there is no consensus on their possible effect on improving the dietary energy and protein utilization, and therefore, on laying hens' performance and mineralization [8]”

L100 What type of experimental design was used?  LSD, CRD, RBD or without blocking?

Authors: Ok. The design was CRD (the animals were randomly distributed in the 4 treatments). We have modified this sentence: “On week 22 of age (day 1 of the experimental period), birds were weighed and randomly distributed in 4 different treatments with 12 replicates/treatment (48 animals/treatment).”

L103, L162 and others: Ti02 – subscript

Authors: Done.

L110 1.07 egg per hen per day?

Authors: Done “…they reached a daily production of 1.07 eggs per hen”.

L115 Please correct reference style.

Authors: Done

L120 Why eggshell weight/percentages were not recorded? It is as essential information regarding eggshell mineralization as shell thickness. What was the idea of measuring albumen thickness and HU? what is the connections of HU or albumen height with Ca, P or phytase supplementation? Moreover, results of these traits are not discussed at all.

Authors: We fully agree with the reviewer. It would have been of interest to have also controlled the weight of the eggshell. Only the thickness of the shell was controlled, which is a good indicator of mineralization, but we have included eggshell weight as a trait to be controlled for future works. Regarding the other variables of egg quality, the usual ones were carried out and those that showed significant differences (the color of the yolk) were discussed.

L134 HU is not measured in %

Authors: Effectively, the Haugh unit value ranges from 0 to 130. Solved.

L135 Please double-check HU equation

Authors: Done: 100 × log10 (H + 7.57 − 1.7W0.37)

Table 1 – please mark which parameters were calculated and which were determined in analyses.

Authors: All the parameters that appears in table 1 were determined with the exception of non-phytate P that was obtained by difference between total P and phytate P (as the superscript indicates).

L172 Were bones crushed to make possible bone marrow removal during boiling/cleaning/degreasing? Cartilage caps were also removed after boiling ?

Authors: Bones were not crushed, but cartilage (soft tssues) was removed after boiling.

L197 Term BIC is more commonly used.

Authors: Done (Bayesian information criterion…)

L208 Please consider removing reporting percentage points changes. You are refereeing to mean/lsmean values, and some traits were determined with an accuracy of percentages

Authors: Done. Changes have been eliminated from the main text following the recommendations of the reviewer.

L231 diet x age

Authors: Done.

L236-243 The description of AID results (Table 3) is insufficient. There is no information about quadric effect for DM. Please also report exact p-values for contrast analysis.

Authors: This sentence has been modified following reviewers recommendation “However, AID of both DM and P with P1000 diet was significantly higher than with NC diet (P<0.001).”

Figure 1  Whishes show SD or SE?

Authors: SE. It has been included in Figure 1: “Bars represent standard errors.”

Table 5 Some data presented in this table should be deleted. It is just multiple presentation of the same results. Only significant changes were observed when ash/Ca/P were expressed in relation to DM, which is a proper presentation of the ash/mineral content in bones.

Authors: We agree with the reviewer. We have reduced the presentation of the data of ash, Ca and P in tibia to % DM.

L317 “8.01 vs. 4.10” units?

Authors: Done. (8.01 vs. 4.10 mg/dL)

L386 the information that "a new 3-phytase" was supplemented in this study is missing in introduction and materials and methods sections. Please provide same characteristic of this new phytase.

Authors: We have eliminated the reference of “a new” as it is not the aim of the study, and this phytase has been already used in previous papers.

Reviewer 2 Report

Line 23   A total of 288 laying hens were used ? 288 or 192 ?

Lines 33-34 change “CTTAD of Ca was significantly higher in P1000 diet than in NC diet at both ages (p<0.001)”  as  “CTTAD of Ca was significantly higher in P1000 diet than in NC diet at 31 weeks of age (p<0.001)”

Line 34 delete “being Ca excretion lower, especially at 31 weeks of age (p<0.001)”  Means with identical superscripts.

2.1 Animals and housing

A total of 288 laying hens (Lohmann brown) of 16 weeks of age were used. …. At arrival, hens were randomly distributed in 72 cages (4 animals/cage) located in two environmentally controlled rooms (36 cages per room)….. and cages were distributed in 4 different treatments with 12 replicates/treatment (48 animals/treatment).  Review the period. Were 288 or 192 hens used?

Lines 110-111 change  “until they reached a production of 1.07 eggs/hen and day.” as  “until they reached a egg production (%) of 1.07.”

Lines 130-132  “On days 52, 53 and 59 of the experimental period, all the eggs laid in the last 24 hours (150 eggs/treatment approximately) were collected for egg quality measurements.” Review the period. The eggs collected seem to me to be a lot with 48 hens per group.

Lines 226-228 change “Respect to mineral utilization, the CTTAD of Ca was significantly higher in P1000 diet than in NC diet at both ages (+1.6 and +3.3 points of percentage at 25 and 31 weeks of age, respectively; P<0.001), “ as “Respect to mineral utilization, the CTTAD of Ca was significantly higher in P1000 diet than in NC diet at 31 weeks of age ( +3.3 points of percentage; P<0.001),

Line 229 delete especially

Line 238 change “Table 3”  as  “Table 2”

Line 243   “+30.1 points”  to check, value too low for a passage from 22.91 to 52.96

Line 315 after minerals in the blood please add reference (Imari, Z,K.; Hassanabadi, A.; Moghaddam, H.N.    Response of broiler chickens to calcium and phosphorus restriction: Effects on growth performance, carcase traits, tibia characteristics and total tract retention of nutrients. Ital. J. Anim. Sci. 2020, 19, 929-939.)

Author Response

Reviewer #2

Line 23   A total of 288 laying hens were used ? 288 or 192 ?

Authors: Effectively, only 192 animals from the initial 288 were selected at the beginning of the experimental period.

Lines 33-34 change “CTTAD of Ca was significantly higher in P1000 diet than in NC diet at both ages (p<0.001)”  as  “CTTAD of Ca was significantly higher in P1000 diet than in NC diet at 31 weeks of age (p<0.001)”

Authors: Done.

Line 34 delete “being Ca excretion lower, especially at 31 weeks of age (p<0.001)”  Means with identical superscripts.

Authors: Done.

2.1 Animals and housing 

A total of 288 laying hens (Lohmann brown) of 16 weeks of age were used. …. At arrival, hens were randomly distributed in 72 cages (4 animals/cage) located in two environmentally controlled rooms (36 cages per room)….. and cages were distributed in 4 different treatments with 12 replicates/treatment (48 animals/treatment).  Review the period. Were 288 or 192 hens used?

Authors: Effectively, only 192 animals from the initial 288 were selected at the beginning of the experimental period. We have improved this information in this section.

Lines 110-111 change  “until they reached a production of 1.07 eggs/hen and day.” as  “until they reached a egg production (%) of 1.07.”

Authors: We prefer to maintain the current sentence. We believe that it is more correct as 1.07 is not a percentage.

Lines 130-132  “On days 52, 53 and 59 of the experimental period, all the eggs laid in the last 24 hours (150 eggs/treatment approximately) were collected for egg quality measurements.” Review the period. The eggs collected seem to me to be a lot with 48 hens per group.

Authors: The values are right. There were controlled 150 eggs/treatment obtained at these days.

Lines 226-228 change “Respect to mineral utilization, the CTTAD of Ca was significantly higher in P1000 diet than in NC diet at both ages (+1.6 and +3.3 points of percentage at 25 and 31 weeks of age, respectively; P<0.001), “ as “Respect to mineral utilization, the CTTAD of Ca was significantly higher in P1000 diet than in NC diet at 31 weeks of age ( +3.3 points of percentage; P<0.001)

Authors: Done.

Line 229 delete especially

Authors: Done.

Line 238 change “Table 3”  as  “Table 2”

Authors: Done.

Line 243   “+30.1 points”  to check, value too low for a passage from 22.91 to 52.96

Authors: As recommended by the other reviewer, it has been removed.

Line 315 after minerals in the blood please add reference (Imari, Z,K.; Hassanabadi, A.; Moghaddam, H.N.    Response of broiler chickens to calcium and phosphorus restriction: Effects on growth performance, carcase traits, tibia characteristics and total tract retention of nutrients. Ital. J. Anim. Sci. 2020, 19, 929-939.)

Authors: Done.

Round 2

Reviewer 1 Report

The Authors have answered all my main issues which were raised during first review. However, the revised version still requires minor clarifications before the manuscript will be suitable for publication.

Statistical analysis: in table 4 and 5 a results of post hoc test are shown for values which, according to presented p-values, do not differ between groups (final bw, fcr, tibia weight, Ca in tibia). In general, post hoc test are performed only when ANOVA shows significant differences between means. Please remove results of post hoc test for data with P> 0.05. Also in materials and methods section the information which post hoc test was used is missing.

L426-427 – This is too categorical statement. As you wrote in the answer to reviewer, you have other data describing eggshell quality (eggshell weight, maybe others) but you decided to not showing this data in present manuscript. Single measurement of eggshell thickens does not provide sufficient grounds for such an unequivocal statement.

L81 mineralization – bone mineralization, eggshell mineralization or both of them?

Author Response

Many thanks to the reviewer for his/her recommendations, below we detail our opinion and changes made:

Statistical analysis: in table 4 and 5 a results of post hoc test are shown for values which, according to presented p-values, do not differ between groups (final bw, fcr, tibia weight, Ca in tibia). In general, post hoc test are performed only when ANOVA shows significant differences between means. Please remove results of post hoc test for data with P> 0.05. Also in materials and methods section the information which post hoc test was used is missing.

Authors: We understand the reviewer's proposal, as it is a fairly widespread view on P-values and the comparison of means. However, we do not completely agree, when in the same model are analyzed different treatments (four in this case) where the effects can be independent (mineral level, phytase level). The aim of ANOVA is to detect whether a factor has a significant effect on a dependent variable globally. An common practice is to pursue multiple comparisons only when the hull hypothesis of homogeneity is rejected. However, in some cases (as in the present work), post-hoc tests can be powerful enough to find significant differences between group means even if the global effect is not significant. Generally, we can consider such results as valid with one exception (protected Fisher LDS). For this reason, the authors prefer to maintain the current presentation of this data in the Tables 4 and 5.

L426-427 – This is too categorical statement. As you wrote in the answer to reviewer, you have other data describing eggshell quality (eggshell weight, maybe others) but you decided to not showing this data in present manuscript. Single measurement of eggshell thickens does not provide sufficient grounds for such an unequivocal statement.

Authors: We agree with the reviewer, we have deleted this statement.

L81 mineralization – bone mineralization, eggshell mineralization or both of them?

Authors: Bone mineralization. We have included the change proposed.